# Phonological processing skills influence text-reading fluency in Russian-speaking adolescents

Tatiana Eremicheva[1,2]*, Yana Nikonova[2], Svetlana Alexeeva[2]

1 Center for Language and Brain, HSE University, Moscow, Russian Federation, 2 Center for Language and Brain, HSE University, Saint Petersburg, Russian Federation

☺ These authors contributed equally to this work.
* teremicheva@hse.ru

## Abstract

It is widely acknowledged that phonological processing skills are crucial for reading acquisition. While most research has focused on examining phonological processing in children, adolescents have remained underexplored. The present study aimed to assess phonological processing (PP) skills in Russian-speaking adolescents using the *Changing Sound in Pseudoword* test from the Russian Test of Phonological Processing (RuToPP). This test involves multiple linguistic processes in its completion and is considered to be the most difficult from the RuToPP. A total of 161 participants (14–18 y.o.) were involved in the study. First, we calculated percentile-based thresholds, which can be used to indicate the potential risk of phonological processing difficulties in adolescents. Then, we analyzed how PP skills influence word, pseudoword, and text reading fluency. The results revealed that PP skills had a small but significant impact on text reading fluency. No significant effects were observed for word and pseudoword fluency. We speculate that, for adolescents, the ability to manage general cognitive load is more crucial for performing complex reading tasks than the level of basic phonological skills.

## Introduction

Phonological processing (PP) skills are one of the most consistently reported predictors of reading [1–3]. They reflect the ability to process spoken and written language using phonological information, i.e., minimally distinguished units of sounds in a language [4]. Properly developed PP skills enable children to successfully acquire reading skills at school [5,6]. This was proved by a plethora of studies conducted mostly on preschool and primary school students [1,3–6]. The results of this research allowed for a timely detection of possible PP deficits, which could negatively affect reading outcomes in further schooling [7,8]. However, some older students continue to experience difficulties with reading, and the reasons for this remain unclear. The knowledge on the exact linguistic and cognitive abilities, which influence reading in

**Data availability statement:** Data are available at the OSF repository (https://osf.io/p6f24/overview).

**Funding:** This research was supported by the Basic Research Program at the National Research University Higher School of Economics (HSE University). The funders had no role in study design, data collection and analysis, decision to publish, or preparation of the manuscript.

**Competing interests:** The authors have declared that no competing interests exist.

adolescents, is limited. Since reading difficulties in younger children are usually tied with PP, we chose to focus on the PP skills in adolescents.

Little attention was paid to the role of PP skills in adolescents' reading. Without a strong body of research on this topic, there was no consensus in the assessment tools appropriate for adolescents. Thus, in this study we use a test that has proven effective in Russian-speaking children, because it was created to measure several linguistic aspects linked to PP skills.

### Phonological processing skills and reading

According to the traditional model reported in [4], there are three components of PP that influence reading acquisition. The first component, phonological awareness (PA), describes one's ability to discriminate and manipulate phonemes of one's language. PA is required to make *phoneme-to-grapheme* correspondence during reading. This skill is extremely important when children are first learning to read, and later when older readers have to process long unfamiliar words. The second component is phonological working memory (PWM), which involves storing the sound-based representations during the decoding process in reading. The third component describes the phonological decoding in lexical access and is often measured with rapid automatized naming (RAN) tasks. This component reflects how fast the sound-based representations (formed in a word) map with its lexical referent, which is essential for effective reading comprehension.

The disruption of one component might result in troubled reading, because of the complex and associated nature of the components [4]. Therefore, to provide a comprehensive assessment of PP, each component should be targeted objectively. The existing research provided various assessment tools, which were used to investigate PP's role in reading. Hence, the PA component was assessed with *phoneme deletion* or *phoneme segmentation* tasks [9,10], the PWM component was assessed using the *digits*, *phonemes* or *nonwords repetition* tasks [11–13], and the last PP component was primarily assessed with the *rapid naming of digits, objects, colors,* or *words* [14–16]. However, the previous findings suggest that the differences in PP tests (e.g., type of stimuli), the age of the participants and reading proficiency lead to inconsistent conclusions about how strongly PP components predict reading.

One factor that might account for these inconsistencies is the complexity level of PP tests. In prior studies it was determined as a number of linguistic and cognitive processes involved in the performance of a PP test [17,18]. Some researchers suggested that it could be taken into consideration while choosing the appropriate PP test. In [7] it was shown that more challenging PP tests from the CTOPP battery [19] predicted reading skills in younger children more effectively. Dorofeeva et al. [18] showed that PP tests involving more linguistic processes predicted better reading performance in primary school children with dyslexia. Therefore, while choosing the most effective PP tool in adolescents, we might consider the complexity of the PP tests as a contributing factor.

The type of reading tests is another important factor that should be considered in reading research. For instance, the evaluation of oral reading fluency in primary

school students was typically conducted using *word* and *pseudoword* reading tests [5,20–22], which require the ability to read separate words or pseudowords without additional context. These tests are appropriate for assessing the early stages of reading skills. These early reading skills serve as a foundation for more comprehensive stages of reading skills (e.g., *text* reading). Moreover, these tests allow the assessment of lexical (with words) and sublexical (with pseudowords) reading routes, proposed by the *Dual Route Cascaded* model [23]. According to this model, reading aloud involves two simultaneous and competing processes, which differentiate in the approach to recoding the written information. Hence, in the lexical reading route, whole units of phonemes are activated simultaneously, leading to the activation of lexical access. In the sublexical reading route, readers perform the gradual decoding of each grapheme into a corresponding phoneme [24]. Therefore, each route involves PP skills on a different level, which justifies the importance of using both tests in research on the PP's role in reading.

To assess more complex reading skills the *text* reading tests were widely used (primarily, in more skilled children) [17,25,26]. These tests require more cognitive processes to accurately read and comprehend the text passages, and allow for a more realistic assessment of reading skills (i.e., since children regularly read texts at school). In order to accurately perform the text reading test, children rely not only on the basic PP skills, but also text reading prosody [27]. Hence, the important idea here is that the choice of reading tests would influence the selection of the appropriate PP tests, since their relationship might differ based on the complexity level of tests. This type of testing instrument is particularly important for adolescents.

## Phonological processing skills and reading in adolescents

Only a few studies previously investigated the relationship between PP and reading in adolescents. The limited amount of research was supported by the idea that PP skills influence reading less in older students [28–29]. However, the precise conclusions cannot be made, since the research on this topic was insufficient. Some researchers found that PP skills predicted reading in older students, but on a different magnitude compared to children [30–32].

In line with studies on children, researchers targeted the traditional PP components while assessing PP skills in adolescents. As for the PA component, when it was assessed with the CTOPP test battery [19] and the *spoonerism* task, it did not predict either silent or oral reading in older students [33,34]. Possibly, the PA had a lack of predictive power in adolescent readers because this component seemed fully automated by the end of primary school and did not require much effort while reading in later years [35]. The PWM component, evaluated in adolescents using the Woodcock-Johnson-III test [36] and the nonword repetition test also did not show any predictive power on oral text reading fluency [30]. This component did not account for a unique variance in adolescents' performance on cognitive and reading tests, suggesting that working memory shared its effect with other skills involved in reading at this age period [30]. However, the tests that were used to solely assess PA and PWM in the observed studies were made for younger children. That is why they might be less sensitive for adolescents and did not show the real impact of PP on reading in this age group. On the contrary, the tests targeting simultaneous activation of multiple PP components might be more efficient in adolescents. Thus, Kairaluoma et al. [31] found that PP consisting of two PA tasks and one PWM task was greatly associated with oral reading speed in adolescents and differentiated in those with and without reading difficulties.

The third component of the traditional PP model remained a consistent reading predictor in adolescent research when measured with RAN [30,31,33]. Unlike in the PA and PWM assessment, the RAN test *simultaneously* involves multiple skills, such as visual processing skills, ability to quickly recode written information (i.e., strings of letters, words, objects) into its phonological form, lexical access, working memory and articulation skills, which might account for RAN's predictive power on reading [4,14,37]. However, it is still not clear, which underlying processes of RAN predict reading better in adolescents and to what extent we can use this test as a tool for assessing PP skills specifically [16].

Therefore, to assess PP skills in adolescents, we need an instrument that will 1) involve assessment of the phonological component of speech processing, 2) include PA and PWM activation, and 3) has a relatively complex structure

requiring multiple cognitive processes in its successful development (as in RAN task). In this study we set out to use precisely this type of the test.

Another key question is what kind of reading test is most appropriate for adolescents. The effect of different reading material was less investigated in older students. Adolescent research focused primarily on the text reading fluency [30,32,38,39]. This is, naturally, the most relevant way to assess reading skills, since in everyday life we read coherent texts. On the contrary, the evaluation of word and pseudoword reading fluency was conducted less frequently and with differing results. For instance, Barth et al. [30] examined how PP skills influence word-level reading in adolescents. They found that phonemic decoding accounted for 15% of a unique variance in word and pseudoword reading fluency. This finding supported the idea to investigate the impact of isolated PP components not only on the text reading fluency, but also on the word and pseudoword reading fluency in adolescents. An investigation of the word-level reading would provide the necessary insights on the basic level of reading skills development, which is crucial for the text-level reading. These insights might highlight the importance of including the basic levels of reading in the clinical assessment and training.

While doing the research on adolescent reading, a contribution of other cognitive skills (apart from PP) and basic participants' characteristics (grade, gender, school) should be considered as well. The reading tasks especially at the text-level require basic processing skills as well as the high intelligence level and well-developed memory skills for a successful comprehension of the text content. Moreover, success in reading depends not only on the cognitive skills, but on the general attitude to reading as well. Those readers, who enjoy reading, tend to have a higher reading speed [40]. It might be especially influential for adolescent reading, since their attitude to reading might change with an increasing complexity of the text passages.

Thus, in the present study we aimed to address all of these issues. Specifically, we (a) chose a phonological test designed to assess PA and PWM and to tap several linguistic operations (similar to RAN, but strictly within the phonological domain); (b) examined its contribution to reading at different levels (text, word, and pseudoword reading); and (c) controlled for a range of cognitive skills and for reading motivation.

## This study

In our study, we assessed PP skills in Russian-speaking adolescents and investigated the ability of PP and other cognitive skills to predict reading fluency in this age group. Moreover, we tested how adolescents' attitude to reading impacts their reading fluency on word-, pseudoword- and text-level reading.

Our first research goal was to conduct a detailed and accurate assessment of PP skills in adolescents using an appropriate testing instrument. We used the most complex test from the Russian Test of Phonological Processing (RuToPP) [17,18]. This test involved five linguistic processes in its performance: (i) input processing, (ii) phonological analysis, (iii) operating with phonemes (targeting the PA component of the traditional PP model), (iv) phonological working memory (targeting the PWM component), and (v) output processing. The chosen PP test was similar to the RAN tasks (used to measure the third PP component) in terms of its complex structure, however, its involved processes were purely phonological (unlike in the RAN tasks). This test was proved to be the most difficult in primary school children [17] and valid for discriminating children with and without dyslexia [18]. Given these advantages and a lack of other specific tools valid for PP assessment in Russian-speaking adolescents, we used this test in the target age group. We also aimed to provide normative data on this PP test for further use in research and practice. In our study we achieved the first goal.

Our second research goal was to examine how PP skills assessed with an effective instrument would influence reading fluency in adolescent students. We compared the effects of PP on three types of reading fluency: word reading fluency, pseudoword reading fluency, and text reading fluency. Examination of word and pseudoword reading was conducted using two tests from the Standardized Test Assessing Reading Skills (STARS) [41]. These tests involved separate sets of stimuli, presented without additional context. Assessment of text reading fluency was conducted using the text that was prepared according to the standardized methodology valid for text reading assessment in Russian-speaking children

(SARS) [42]. This text was selected by professional speech therapists and was standardized for adolescents with Russian as their native language [43]. In our study we achieved the second goal.

Our first hypothesis was that PP skills assessed with the phonological test from the RuToPP battery would influence text reading fluency, since both the PP and text reading test that we chose require multiple cognitive skills in their completion. It was previously stated that PP tests involving more cognitive skills were more challenging and, therefore, obtained higher predictive power for reading skills in adolescents. Our second hypothesis was that for pseudoword reading fluency, PP skills would have more explanation than for word reading fluency. It was supported by the idea that in pseudoword reading, PP skills were involved more actively, while for word reading, other processing skills were required more [44,45].

Additionally, we assessed the nonverbal intelligence level with Raven's Standard Progressive Matrices and collected information on demographic features, memory skills and attitude to reading with a detailed questionnaire. We tested whether these additional skills had an impact on word, pseudoword and text reading fluency in adolescents. We hypothesized that PP skills might uniquely account for the variability in adolescent reading. The other cognitive skills (nonverbal intelligence, memory skills) and attitude to reading would share variance with other factors in the model. The simultaneous assessment of all these reading predictors was not conducted previously, therefore, this part of the study will be rather exploratory.

We decided to study adolescents with Russian as their native language. Russian is a language with a semi-transparent orthography. The correspondence of graphemes to phonemes can be made easily [46], whereas the correspondence of phonemes to graphemes requires extended knowledge of orthographic and morphological rules [47]. Therefore, Russian speakers engage PP skills in reading differently (depending on the reading task), which attracts researchers' attention. The relationship between PP and other cognitive skills with reading is majorly understudied in Russian speakers. The existing studies were mostly conducted in Russian-speaking preschool and primary school students [9,48]. In Russian-speaking adolescents, researchers have studied the effects of grammatical processing, while the influence of PP on reading has not been studied yet [49–54]. Thus, we decided to conduct the present study on adolescent native speakers of Russian.

## Methods

### Participants

The study included 179 typically developing Russian-speaking adolescents. All participants were students in grades 8–11 (M = 15.8 years, SD = 1.54) attending regular schools, gymnasiums, or lyceums. We chose the schools for participation based on the agreement with their authorities. Participants were recruited voluntarily, after they received an invitation to join the experiment at school. The sample comprised 102 girls and 77 boys. The data collection was started on the 25th of October 2022 and was finished on the 4th of October 2024. Parents or legal guardians confirmed that none of the participants had been diagnosed with visual or hearing impairments, neurological disorders, or learning disabilities. Written consent forms were obtained from both participants and their parents or legal guardians. The study was first approved (Registration Card No. 115-03-18; Protocol No. 115-02-7) and then prolonged (Registration Card No. 115-03-20; Protocol No. 115-02-5) by the Ethics Committee of Saint Petersburg State University.

In our study we aimed to include the results of typically developing participants, who had no history of diagnosed impairments. Therefore, one of our inclusion criteria was the nonverbal intelligence level within the age norms. To assess nonverbal intelligence, we used Raven's Standard Progressive Matrices [55] with the age norms for Russian-speaking individuals [56]. Participants with the scores below the age norms were excluded from the analysis (16 participants). The exclusion of the participants with lower levels of nonverbal intelligence is a standard procedure for a study with typically developing participants.

Another inclusion criteria was the availability of all the tests from the experiment procedure. Hence, we excluded the results of two participants who did not understand the instructions for the word and pseudoword reading tests. The final sample included 161 participants. Detailed information about the participants is presented in Table 1.

**Materials**

**Word and pseudoword reading tests.** Reading skills on a word level were assessed with the Standardized Test Assessing Reading Skills (STARS) [41]. The test battery consisted of four lists (two with words and two with pseudowords). In our study, we randomly chose list C (with words) and list D (with pseudowords), since lists C and D are the parallel versions of lists A and B. The authors of the test suggest using one pair of the lists in the experiment and the second pair to use as a retest measure when necessary.

Each list included 144 stimuli, which were presented in order of increasing linguistic complexity. The complexity was determined by the level of word frequency, the number of letters, the number of syllables and syllable structure. The morphological features were not included in the criteria determining complexity level of the stimuli. Therefore, more frequent stimuli, with fewer syllables of less complexity, were presented first, followed by stimuli with decreased frequency, containing more syllables with more complex structure. List C, with words, included nouns, adjectives, and verbs from the Russian language. They were chosen from the open database for Russian language [57] and were ranged by the age of acquisition. The frequency level of words was determined using the open database of stimuli in Russian [58].

List D included pseudowords which were created by taking existing Russian words from list C and replacing 1–3 graphemes in them. One grapheme was changed only in short words with one syllable, whereas in longer words (with two or more syllables) more graphemes were changed. For example, the existing Russian word "поле" [ˈpolʲə] was turned into the pseudoword "фоме" [fomʲə] by replacing two graphemes. Here, vowels were replaced only with vowels and consonants were replaced only with other consonants. This allowed to keep the length and syllable structure of the stimuli. The phonotactic legality in the pseudowords was consistent with the phonetic rules in Russian, which is beneficial for the PP assessment. The resultant pseudowords were placed on the list in the same order as the original words were presented.

**Text reading test.** Text reading skills were evaluated using the text called *The Caring Flower.* We chose the text used in the study [43]. They created a text based on a narrative by Konstantin Paustovsky. The original text, which was suggested by speech therapists, is not included in the standard school curriculum. Using a text unfamiliar to participants helped minimize prior exposure and thus increase the reliability of the results. The final version of the text consisted of 181 words and five paragraphs. An average lemma frequency was 452.7 occurrences per million words with an average word length of 6.8 characters.

After we assessed text reading fluency, we measured reading comprehension to ensure that all participants read the text thoroughly. To evaluate the comprehension skills, we provided eight questions related to the text. The list of questions included both close-ended questions, i.e., those that require "yes"/"no" answers, and open-ended questions formulated according to the text content, i.e., "What color are the flowers of the cypress?". Prior to the assessment of reading comprehension, we also formulated a list of the corresponding keys to these questions. This was done to increase inter-rater

**Table 1. Information about participants included in the analysis.**

| Grade | 8th | 9th | 10th | 11th | Total |
|---|---|---|---|---|---|
| Sample (F/M) | 41(20/21) | 55(33/22) | 35(19/16) | 30(14/16) | 161(86/75) |
| Mean age (years; months) and SD (months) | 14; 7 (6) | 15; 7 (6) | 16; 5 (5) | 17; 4 (5) | 15; 10 (12) |

reliability. Estimates of the inter-rater reliability for reading fluency and comprehension are provided in the Manual Scoring section of this manuscript.

The characteristics of the test we used were inspired by the Standardized Assessment of Reading Skills (SARS) [42], which is one of the most common instruments to assess text reading skills in Russian. This test is highly familiar among Russian children as it is widely used for different research and diagnostic purposes. We decided to use a new test to avoid possible familiarity with the SARS materials.

**Phonological test.** Phonological processing skills were assessed using the *Changing Sound in Pseudoword* (CSP) test from the standardized phonological test battery, i.e., the Russian Test of Phonological Processing (RuToPP) [17,18]. The RuToPP contained seven phonological tests varying in linguistic complexity. The linguistic complexity was determined by the number of linguistic processes required for the successful completion of each test. The tests involving two speech processes were defined as less complex, and the tests with five speech processes were the most complex ones. The detailed information on each test from the RuToPP test battery can be seen in [17,18].

In our study, we decided to use one of the most complex tests, namely the *Changing Sound in Pseudoword* (CSP) test. This test involved five speech processes in its completion: 1) input processing (speech perception, phonological decoding); 2) phonological working memory; 3) phonological analysis; 4) operations with phoneme sequences; 5) output processing (phonological retrieval and articulation). In this test, the participants were asked to change a given phoneme in a pseudoword and name the new pseudoword with the changed phoneme. For example, the instructor would say: "Replace sound /b/ with /p/" and then continue with the pseudoword "nuba". The participant would have to answer "nupa". This test consisted of 24 stimuli.

In the past, this test was found to be one of the most sensitive tests from the RuToPP battery for predicting reading difficulties in primary school students [18]. Previous studies showed that 7–11-year-old children had the lowest scores on this test [17]. That is why we found this test appropriate for PP skills assessment in adolescent students, since we expected that we would face the ceiling effect in the results of the other six tests from the RuToPP battery.

## Questionnaire

The participants also completed a questionnaire collecting demographic information, such as gender, grade, date of birth, and school name. In addition, the questionnaire included items related to memory functioning. We asked whether the participants experienced difficulties recalling words, names, or dates, and whether they had trouble remembering instructions presented orally or on paper. Each positive response was scored as 1 point. The *total memory score* was obtained by summarizing the points on all relevant items. Participants also described their attitude toward reading using a 5-point Likert scale (1 = lowest, 5 = highest) and reported how frequently they read for pleasure. The results on these scales were summarized and included in a composite *reading habits index*.

## Procedure

All participants were tested individually in a quiet room. The data was collected as part of a bigger project, containing more tests in the experiment session. The tests that we included in this study were used in the following order. The participants completed the questionnaire (2–3 minutes). Then they performed the Raven's Standard Progressive Matrices test, which took approximately 15–20 minutes. Then they had a short break. After the break, the participants performed the word reading test (1–1,5 minutes) followed by the pseudoword reading test (1–1,5 minutes) and the text reading test (3–5 minutes). Finally, they completed the CSP test, which lasted 5–7 minutes. The order of the tests was consistent for all participating adolescents due to such a complex procedure.

**Raven's standard progressive matrices.** The assessment was conducted using the computer version of the test built with PsychoPy [59], with a 20-minute time limit. Before the start of the test, task instructions were presented on the screen, and the participants could ask the experimenter for clarification if needed. During the test, the participants saw pictures from the original test, with six (in some pictures – eight) possible answers provided in the lower part of the screen. The participants were asked to press a key corresponding to the correct number on the keyboard (1–6 or 1–8).

The accuracy scores in this test were calculated automatically for each participant.

**Word and pseudoword reading test.** Before the start of the assessment, the participants were asked to read the stimuli as quickly as possible without making errors. With the prior consent from the participants, we recorded audio of them reading the stimuli from the STARS test battery. Further, it was manually scored for each participant. We calculated word and pseudoword reading fluency as a number of correctly read stimuli during the first minute of reading (in line with the methodology described in [41]).

**Text reading test.** The participants were instructed to read the text as quickly as possible without making errors. The experimenter also clarified that, upon completing the reading task, they would be required to answer the comprehension questions about the text's content. With the prior consent, we recorded 1 minute and 20 seconds of text reading, which was further scored manually. After reading the text, participants answered eight comprehension questions. If necessary, the experimenter repeated a question. The participants also had an option to respond with "I don't know" or "I don't remember."

**Phonological test.** The phonological (CSP) test was presented using the pre-recorded audio run from the experimenter's computer. In the beginning, the participants completed three training trials, which were not scored. The responses to the subsequent 24 stimuli were recorded. After data collection, all recordings were manually transcribed, and accuracy scores were calculated.

## Manual scoring

The data was gathered by six testers. All of them received the same instruction on data gathering and had the same keys to the tests. When the testers finished the data gathering, they did the manual scoring of the results in the four tests: word, pseudoword and text reading tests and the CSP test.

The answers in these tests were audio recorded during the testing session. The testers listened to the recorded data and scored the number of the correct answers. In the reading tests, mispronunciation, omission or adding of the words were counted as errors. Incorrect stress placement was not counted as an error according to the guidelines of the authors. In the CSP test, mispronunciation of the target answer was counted as an error. Self-corrections were not counted as errors in any test. The testers evaluated only the final pronunciation of the stimuli. Each test was evaluated by two testers and the final decision was made by a third tester.

After the scoring was finished, we calculated the inter-rater reliability. To do this, we calculated the Cohen's kappa coefficient [60] using the function *cohen.kappa* from the *psych* R-package [61]. The level of inter-rater reliability was substantial for the accuracy in the CSP test ($k = 0.69$). For the word, pseudowords and text reading fluency (measured as number of correctly read words/pseudowords per minute) inter-rater reliability was calculated using the *icc* function from the *irr* package in R [62]. Results indicated an excellent inter-rater reliability with $ICC(A,1) = 0.999$ for word reading fluency, $ICC(A,1) = 0.974$ for pseudoword reading fluency and $ICC(A,1) = 0.999$ for text reading fluency. As for the comprehension – the level of agreement was excellent ($k = 0.89$).

## Data analysis

The whole data-analysis was performed in the R environment [63]. First, we calculated descriptive statistics and provided normative scores on the PP test. We provided the means, standard deviations, and percentile-based thresholds for four different grades and for all participants from grades 8–11 in total.

Our next step was to analyze whether the phonological processing skills predict word, pseudoword and text reading fluency in adolescents using (mixed-effect) linear modelling (see below). The dependent variables were the number of correctly read words per minute in the word reading test, the number of correctly read pseudowords per minute in the pseudoword reading test, and the number of correctly read words per minute in the text reading test. The main independent variable was the CSP test scores. Three independent analyses were conducted.

The control predictors were participant's grade, gender (provided in the demographic questionnaires), the Raven's Standard Progressive Matrices scores, the total memory scores, the reading habits index, and the comprehension question scores. Before performing statistical analysis, normality of distributions were tested using Shapiro–Wilk test (*shapiro.test* function from *stats* package in R) [63] as well as density plots from *ggplot2* package [64] (see Fig 1 and S1 Fig).

To analyse the relationship between each of the dependent variables and the mentioned predictors we at first run a linear mixed effect model using the *lmer* function from the *lme4* package in R [63].

In these models, we included all independent and control variables along with their interactions with the CSP test score, as well as a random intercept for school. Subsequently, we applied the *step* function from the *stats* R-package [63], which performs a stepwise model selection by sequentially removing the non-significant predictors. The random effect for school caused singularity issues in some of the models (see below), so it was excluded from the final analysis. In such cases we built linear regression models using the *lm* function from the *stats* package in R [63]. We first fitted a full model including all previously listed main and control predictors and their two-way interactions with CSP test score, and then applied the *step* function to select the final model. The phonological processing score was retained in the final models (see the *Results* section below) regardless of its significance, as it was the main focus of the study.

Since the CSP test scores were not normally distributed (Fig 1), we applied a logarithmic transformation to the test results and used a logarithmic score on the CSP test as a predictor. The grade was included as a numerical variable, and gender variable (a factor) was coded using an orthogonal contrast specified with the *contrasts()* function from the *stats* package in R (–0.5/ 0.5 effect coding) with female as the reference level. All numeric variables were mean-centered, so that their values have a mean of zero. Plots were fitted using the *ggplot2* package [64]. Statistical significance was determined using an alpha level of 0.05. We calculated partial $\eta^2$ to estimate effect size for the predictors in the final models using the *eta_squared* function from the *effectsize* R package [65].

## Results

**Thresholds for the phonological test.** The first step of the analysis was to calculate thresholds for the CSP test indicating potential risk of phonological difficulties in adolescents. The score distribution (see Fig 1, Table 2) demonstrated that more than a half of the participants achieved an accuracy level above 95% (104 out of 161 participants, 64.6%), and that the data was not normally distributed. Therefore, we calculated percentile-based thresholds, which seemed more appropriate to use. We chose the 10%-cutoff for indicating the participants at risk based on the standards provided in [66]. The calculated thresholds are presented in Table 2. Since the number of the participants in each class varied from 31 to 55, we propose that these thresholds are sample-referenced and exploratory. According to the calculated thresholds, those students of the 8th grade, who score less than 83%, and those students of the 9th grade, who score less than 85% in the PP test, potentially may have phonological difficulties. Those students of the 10th and 11th grades, who score less than 88%, also can be at risk for phonological difficulties.

**Word reading fluency depending on phonological processing skills.** Our second step was to analyse whether PP skills influence word reading fluency. The final model was a linear regression model with the number of correctly read words per minute as the dependent variable, and the log-transformed CSP test scores and gender as the independent variables. Other predictors were not retained in the final model, as the *step* function excluded them during the stepwise

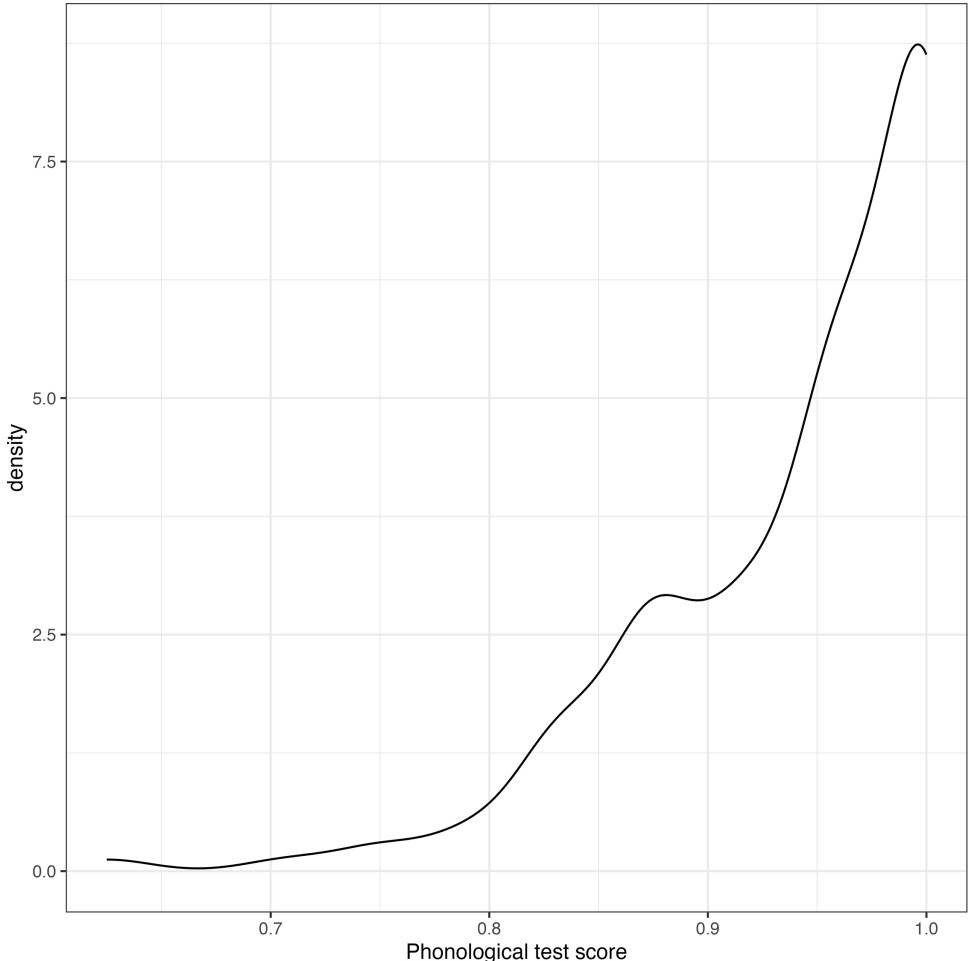

**Fig 1. Distribution of the phonological test scores.**

**Table 2. Percentile-based thresholds indicating potential risk of phonological difficulties.**

| Grade | N | 5% | 7% | 10% (risk) | 15% | 25% | 50% (median) | 75% | 85% | 90% | 93% | 95% |
|-------|------|------|------|------------|------|------|--------------|------|------|------|------|------|
| 8 | 41 | 0.83 | 0.83 | 0.83 | 0.83 | 0.88 | 0.96 | 1.00 | 1.00 | 1.00 | 1.00 | 1.00 |
| 9 | 55 | 0.83 | 0.83 | 0.85 | 0.88 | 0.92 | 1.00 | 1.00 | 1.00 | 1.00 | 1.00 | 1.00 |
| 10 | 35 | 0.83 | 0.85 | 0.88 | 0.88 | 0.90 | 0.96 | 1.00 | 1.00 | 1.00 | 1.00 | 1.00 |
| 11 | 30 | 0.81 | 0.88 | 0.88 | 0.89 | 0.92 | 0.96 | 1.00 | 1.00 | 1.00 | 1.00 | 1.00 |
| Total | 161 | 0.83 | 0.83 | 0.83 | 0.88 | 0.92 | 0.96 | 1.00 | 1.00 | 1.00 | 1.00 | 1.00 |

N, Number of participants; M, Mean phonological test score; SD, Standard deviation.

selection procedure. The model (see Table 3) was statistically significant ($F_{(2,158)} = 7.185$, $p < 0.001$, Adjusted $R2 = 0.07$). However, we did not find any significant effect of the phonological processing skills on word reading fluency ($p = 0.46$). We found that gender significantly affected word reading fluency ($p = 0.001$). The analysis showed that the female participants read words significantly faster than the male participants. Results indicated a small effect of phonological processing

(partial η²=0.01) and a medium effect of gender (partial η²=0.07) on word reading fluency. The plot for the model is presented in Fig 2, the output of the model is reported in Table 3.

**Pseudoword reading fluency depending on phonological processing skills.** In the third step of the analysis we investigated the effects of PP skills on the pseudoword reading fluency. The final model included the number of correctly read pseudowords per minute as the dependent variable, the log-transformed CSP test scores as the independent variable, and random effects for school. In this part of the analysis, we did not find significant effects of the phonological processing skills ($p=0.09$, $β=22.79$) (see Fig 3) on the pseudoword reading fluency (see Table 3). Results indicated small effect size for the log-transformed CSP test scores (partial η2=0.02).

**Text reading fluency depending on phonological processing skills.** The last step of the analysis was to investigate whether adolescents' performance in the CSP test can predict text reading fluency. The final model included the number of words read correctly per minute as the dependent variable, the log-transformed CSP test scores and the reading habits index as the independent variables. We didn't add random effects in this model due to

**Table 3. Results of the linear regression models.**

| Predictors | Word reading test | | | Pseudoword reading test | | | Text reading test | | |
|---|---|---|---|---|---|---|---|---|---|
| | Estimate (β) | 95% CI | p-value | Estimate (β) | 95% CI | p-value | Estimate (β) | 95% CI | p-value |
| (Intercept) | 104.39 | 101.03–107.76 | <0.001 | 62.47 | 59.10–65.83 | <0.001 | 147.74 | 143.01–152.46 | <0.001 |
| Phonological test score (log-transformed) | 12.55 | −21.25–46.36 | 0.464 | 22.79 | −3.63–49.22 | 0.090 | 47.15 | −0.38–94.68 | 0.052 |
| Gender | −9.41 | −14.70 − −4.13 | 0.001 | | | | | | |
| Reading habits | | | | | | | 2.36 | 0.66–4.06 | 0.007 |
| Random Effects | | | | | | | | | |
| $σ^2$ | | | | 165.46 | | | | | |
| $τ_{00}$ | | | | 11.20 $_{school}$ | | | | | |
| ICC | | | | 0.06 | | | | | |
| N | | | | 26 $_{school}$ | | | | | |
| Observations | 161 | | | 161 | | | 161 | | |
| $R^2$/ $R^2$ adjusted | 0.083/ 0.072 | | | 0.018/ 0.080 | | | 0.076/ 0.065 | | |

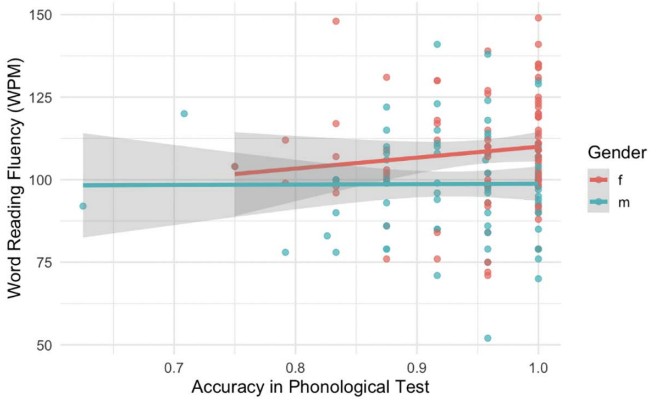

**Fig 2. PP test score versus word reading fluency with regression line superimposed.**

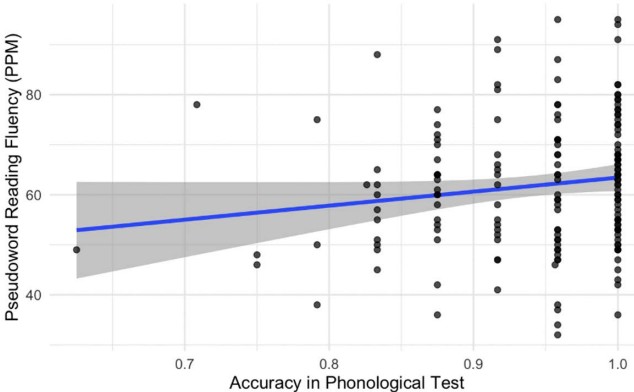

**Fig 3. PP test score versus pseudoword reading fluency with regression line superimposed.**

the singularity issues. The model was significant ($F$ (2,158) = 6.523, $p$ = 0.002, Adjusted R2 = 0.065). The results are presented in Fig 4.

We found that the reading habits index significantly influenced text reading fluency (see Table 3), indicating that the better reading attitude led to the faster reading ($p$ = 0.007). As for the CSP test scores, the significance level was $p$ = 0.052. We compared the model including the CSP test scores (Adjusted R2 = 0.065) with the model without the CSP test scores (Adjusted R2 = 0.048). We found that this predictor explained around 1.7% of the additional variance. Phonological processing showed a small effect on text reading fluency (partial η2 = 0.03), while reading attitude demonstrated a medium effect (partial η2 = 0.05).

## Discussion

In our study, we investigated phonological processing skills and reading in adolescents. We provided a detailed assessment of PP skills and reading skills on two levels, i.e., word- and text-level reading. To the best of our knowledge, this study was the first to simultaneously investigate the effects of PP skills on word, pseudoword and text reading fluency. Moreover, we investigated the effects of other cognitive skills, attitudes to reading in adolescents and several basic

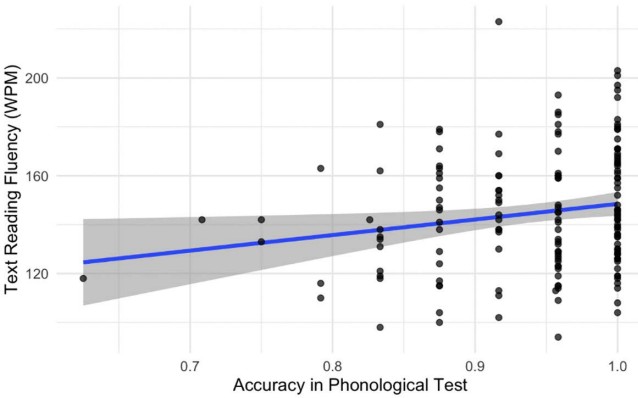

**Fig 4. PP test score versus text reading fluency with regression line superimposed.**

demographic features. The implications from our study are important for further research in adolescents' reading difficulties [30,34,38] and are potentially beneficial for clinical practice.

**Thresholds for the phonological test.** To provide information beneficial for practice, we conducted a systematic assessment of PP skills in adolescents and calculated percentile-based thresholds for this PP test as our first research goal. We conducted the assessment of PP skills with the most complex phonological test (*Changing Sound in Pseudoword*; CSP) from the RuToPP test battery [17,18]. The thresholds on the CSP test were calculated for each grade (8–11) and for all participants (163). The number of the participants in each grade was less representative to derive the grade-specific normatives appropriate for practice. The results in the CSP test did not differ significantly in the participants from different grades, therefore, we suggest to rely on the thresholds calculated for all participants.

Though the resultant thresholds are sample-referenced and rather exploratory, they still might be beneficial for research and practice. This is the first normative data on the standardized phonological test for Russian-speaking adolescents, calculated on a sample of 163 participants. The data was collected in several schools, ranging from the public schools in small towns to the lyceums and gymnasiums in a big city. Therefore, the thresholds were calculated including the results of the participants with different socio-economic statuses. Previously, normative scores for PP tests were calculated only in younger Russian-speaking children [67]. When analyzing PP skills in Russian-speaking adolescents, a lot of practitioners relied on their personal experience, limited to the patients they had met during their clinical practice. We propose that our thresholds might be applicable to research and clinics, since they were calculated on the balanced sample from different schools, and the thresholds were based on the standardized PP test. However, the reported thresholds should be used cautiously, with attention to the limitations of our sample.

**Phonological test and text-level reading.** As our second research goal, we investigated how PP skills influence reading in Russian-speaking adolescents. Along with PP assessment, we used three tests to assess reading performance: word reading test, pseudoword reading test, and a text reading test. Altogether, these tests were used for a systematic assessment of reading skills in adolescents and for comparing the influence of PP skills on different levels of reading, i.e., word- and text-level. First, we hypothesized that PP skills in adolescents would predict their text reading fluency. The results of the regression analysis showed that PP skills were associated with text reading fluency on a small level of significance and should be acknowledged. This is partially consistent with the previous studies. Our PP test targeted PA and PWM components of the traditional PP model (see the *Introduction* section), whereas most of the studies showed no significant effect of these components on text reading skills in adolescents. Instead, the RAN test was proved to be a consistent predictor of the text reading skills in this age group [30–32]. We may have found the similar effect in our phonological test due to its similarity to the RAN in terms of its complex structure: both our PP test and RAN involve multiple cognitive processes for successful completion. Therefore, the question on what makes the RAN a significant predictor in children and adolescents remains unanswered. We propose that the number of cognitive processes and their combination can be a reliable source of its predictive power. Thus, further investigations on reading predictors in adolescents might include tests with a complex structure (requiring multiple cognitive and linguistic skills for their completion).

**Phonological test and word-level reading.** As for our second hypothesis, we proposed that PP skills would have a more significant effect on the pseudoword reading fluency than on the word reading fluency. This idea was motivated by the features of a pseudoword reading test. To complete this task, participants needed to decode unknown pseudowords by comparing each grapheme to existing phonemes. This way, PP skills were involved in the completion of the pseudoword reading task more than in word reading, where participants extracted the meaning of the words from their lexicon. Thus, they relied less on decoding skills and more on their verbal knowledge. The results of our analysis showed that there were no significant effects of the CSP test on both types of word-level reading. Therefore, our second hypothesis was not confirmed.

We suggest that this finding can be accounted by three reasons: 1) the sample size in our study was not sufficient and requires additional recruitment of the participants; 2) the CSP test is not sensitive enough for this age group: therefore, it

does not show the full picture of PP skills development in adolescents; 3) PP skills in adolescents do not predict reading on a word-level. Regarding the first reason, it was shown that some studies recruited smaller samples [34] and did not find any significant effects of PP skills on word-level reading as well. However, studies that used bigger samples (near 2000 students) presented similar findings: PP skills were not significant predictors of word and pseudoword reading skills [30]. Hence, the matter of sample size might not be the most important one. The second issue, addressing the sensitivity of our PP test, can also be disputed. We found variance in the results of our participants: although many students scored above 90% in the phonological test, some got lower scores (see Figs 2–3). Thus, we managed to assess PP skills in Russian-speaking adolescents in a unified and standardized way. Therefore, we consider the third conclusion that has influenced our results: PP skills are no longer significant in predicting reading fluency on a word-level in adolescents. This finding is consistent with the previous studies [31,34], where phonological tests did not influence neither word nor pseudoword reading fluency. It is possible due to the higher levels of development reached by both PP skills and reading skills in later years at school. The formation period of these skills is set to be finished in adolescence. Therefore, the dependence level decreases over the years [35], because most of the typically developing adolescents successfully complete word and pseudoword reading tests (i.e., there is no variation to predict). In the previous study Barth et al. [30] found that the PP skills significantly predicted word-level reading. We suggest that this finding might be evident because authors used word attack and word identification tests from the Woodcock reading mastery test revised [68]. These tests assess other components of reading itself (i.e., word and pseudoword reading *accuracy*), rather than the PP skills. Still we advise to continue investigation of word-level reading in adolescents, but with an increasing complexity level of stimuli.

As a side effect, we found that female participants read words significantly faster than male participants did. These gender differences in reading were not confirmed for pseudoword and text reading fluency. The results provided by the previous studies are partially consistent with ours. In general, it was shown that girls read significantly faster than boys [69,70]. This was also supported by the neurophysiological studies where girls showed more efficient neural processing of reading tasks doing fMRI, which correlates with faster word recognition [71]. However, there was evidence supporting the idea that boys were faster readers than girls when they outperformed girls in the word reading test specifically [72]. There is a possibility that the reported inconsistencies on word-level reading and gender also might be due to the differences in the transparency level of Russian and English orthographies. Furthermore, our results are consistent with previous findings on child word learning [73]. In this study, girls showed higher performance in learning phonologically familiar words, which can relate to word reading fluency, possibly reflecting a more efficient use of the lexical route for recognizing familiar words. This advantage disappears for phonologically unfamiliar or novel words, which rely more on sublexical decoding strategies rather than lexical access.

**Other cognitive skills and attitude to reading.** Additionally, we investigated the role of other cognitive skills (apart from PP) and attitude to reading in adolescents in the explorative fashion. We hypothesized that the nonverbal intelligence level, memory skills, attitude to reading and comprehension skills would have shared effects on the word, pseudoword and text reading fluency. We found that only attitude to reading was a significant predictor of the text reading fluency, whereas it did not show significant effect for word and pseudoword reading fluency. Those who enjoyed reading, read the text faster. This is in line with the previous studies, proving that the attitude to reading is strongly associated with the results in reading tests [40]. The non-significant effect of nonverbal intelligence is also consistent with the previous findings in adolescents, where the nonverbal cognition did not account for a variance in text reading results [30]. We suggest that our findings on memory and comprehension skills require additional testing, since we used less reliable instruments for assessing these skills (i.e., questionnaires instead of behavioral tests).

**Limitations of our study and directions for further research.** One of the limitations of our study is the ceiling effect in the PP test results. According to Table 2, we can see that almost 75% of our participants scored the highest in the chosen CSP test (regardless of the participants' grade). Therefore, we suppose that the CSP test is not challenging enough for adolescent participants. We chose this test since it was the most complex one from the standardized test

battery for Russian-speaking children [17,18]. This test involved five linguistic processes specifically in the phonological domain. There were no corresponding tests in Russian that we could successfully implement in our study to assess PP skills in adolescents. Therefore, this test was the best option for our research goals, especially considering the fact that some of our participants still made mistakes while performing the CSP test. Hence, the most challenging were the probes where the participants had to change the palatalized consonants (i.e., the consonants that are pronounced with the tip of the tongue raised to the hard palate). In Russian, palatalization is a distinguishing phonetic feature that determines a word's meaning. In the CSP test, the participants had to change the sound /v'/ to /v/ in a pseudoword /v'a/ and name a new pseudoword /va/. In the probes with this change, the participants gained around 85% on average. The same was true for those probes, where the participants had to change /k/ to /k'/ (e.g., in a pseudoword /ka/); the participants also scored around 85% in these probes. Therefore, a close examination of the responses to different stimuli showed that operating with palatalized sounds was more difficult for adolescents, which proved that not all aspects of the PP skills are equally developed at this age, with some still requiring more effort to execute.

Previous research, which was conducted for other languages, relied on the following tests to assess PP skills: subtests from the CTOPP test battery [19], rapid naming tasks [30], spoonerism task [34], etc. Hence, in further studies, we should try to adapt the following tests to Russian or develop a new, appropriate tool. Moreover, the existing tools can be modified by adding more challenging stimuli. For instance, the CSP test can be developed including longer pseudowords with less obvious combinations of phonemes. This will increase the complexity level of the test, which is relevant for PP assessment in adolescents. We suggest continuing the investigation of the relationship between PP skills and reading after additional assessment with more tests. Furthermore, additional aspects of PP skills can be included. In this study we measured the accuracy in the PP tests and used this result as a reading predictor. However, in future studies the response latency in phonological tests might be used as a separate measure. It will not only allow us to assess the PP skills but will indicate the level of cognitive load while performing the PP tasks. Thus, it might be a reliable measure in adolescents. The assessment of PP, other cognitive skills, and reading in adolescents might be conducted in the longitudinal form of study, which will also add valuable information on the skills development.

Another limitation of this study was the small effect size of PP on reading. Our analysis showed that the model with text reading fluency included PP as a predictor. The analysis on text-level reading revealed that PP skills in Russian-speaking adolescents accounted for 1.7% of the variance in their text reading fluency. On the contrary, we found that phonological decoding in English-speaking adolescents accounted for up to 10% of the variance in oral reading fluency [30]. We suppose that these differences in the effect size can be explained by language-specific features. Although Russian has, on average, longer words, its orthography is more transparent than English. It was proved that the PP skills in languages with more transparent orthographies have a smaller impact on reading [10,74–76]. Hence, the smaller effect size of PP might be evident due to the faster automatization of orthographic skills in Russian speakers. Probably, Russian-speaking adolescents fully automate the correspondence of graphemes to phonemes at 8–11 grades. Therefore, they might rely on the PP skills less during reading (even when reading averagely longer words). However, the effect of PP on text-level reading is still evident in this age group and it resists the stepwise model selection. Considering these arguments, we suggest that the PP's role in adolescent reading deserves attention, but including broader cognitive skills.

The possible order effect of the tests we used also limited the reliability of the reported findings. The protocol of the test session included more tests than we reported in this study (since this study is part of a bigger research project). Therefore, the order of all tests was fixed and included short breaks. This allowed us to minimize the inconsistency between the testers who collected the data. Moreover, the possible fatigue or carryover effects were easier to control in the fixed order of the tests, since they were evident approximately at the same period (i.e., during the completion of nearly the same tests) in all children. This allowed us to compare the results of the participated adolescents, since they were in the same conditions (unlike in the situation with the counterbalanced order of tests).

Overall, the relationship between PP skills and reading in adolescents should be studied in more detail. Differences in methodological aspects of assessment provide controversial results and produce gaps for further research. Conducting more research on adolescent students would provide a better understanding of how different cognitive processes change over time. Furthermore, it will be useful for clinical practice, since there is a lack of theoretical implications for this age group.

## Conclusions

In this study, we examined phonological processing skills in Russian-speaking adolescents and how PP skills influence reading skills. We conducted a comprehensive assessment of PP skills using the most complex test from the RuToPP test battery [17]. We evaluated reading skills in three dimensions: word reading, pseudoword reading and text reading. This detailed assessment of reading skills allowed us to test how PP skills influence reading on different levels, i.e., on a basic word-level and on a more complex text-level. As a result, we found that PP skills had a significant but small effect on text-level reading, but not on word-level reading. This finding shows that PP skills can still be engaged in reading even in more adult students of the 8–11th grades. However, the complex structure of our PP test suggests that it is not specific phonological skills, but rather the overall level of cognitive load that may significantly influence adolescents' text (but not word or pseudoword) reading fluency. Therefore, further studies might be devoted to the assessment of cognitive load during test completion in adolescents.

## Supporting information

**S1 Fig.  The distribution of cognitive and reading measures.**
(TIF)

**S2 Fig.  Word reading fluency scores depending on the participants' gender, grade, cognitive and reading measures.**
(TIF)

**S3 Fig.  Pseudoword reading fluency scores depending on the participants' gender, grade, cognitive and reading measures.**
(TIF)

**S4 Fig.  Text reading fluency scores depending on the participants' gender, grade, cognitive and reading measures.**
(TIF)

**S1 Table.  Descriptive statistics for cognitive and reading tests.**
(PDF)

## Acknowledgments

We gratefully acknowledge Viktoria Koltuntseva, Alexandra Burdyna, Alexandra Cherevik and Alisa Lezina for performing data collection; Alexandra Cherevik and Timofey Dremin for helping with data transcribing. We are also indebted to the administration and teaching staff of the institutions that provided facilities for conducting our research, namely the Laboratory of Continuous Mathematical Education (LNMO), Secondary School No. 65 with Advanced Study of French in the Vyborgsky District of Saint Petersburg, Secondary School No. 494 in the Vyborgsky District of Saint Petersburg, and Gymnasium No. 11 in the Vasileostrovsky District of Saint Petersburg. We extend our sincere thanks to all our participants and their parents or legal guardians for their involvement in the study.

## Author contributions

**Conceptualization:** Tatiana Eremicheva, Yana Nikonova, Svetlana Alexeeva.

**Data curation:** Yana Nikonova, Svetlana Alexeeva.

**Formal analysis:** Tatiana Eremicheva, Yana Nikonova, Svetlana Alexeeva.

**Methodology:** Svetlana Alexeeva.

**Project administration:** Svetlana Alexeeva.

**Resources:** Svetlana Alexeeva.

**Writing – original draft:** Tatiana Eremicheva, Yana Nikonova.

**Writing – review & editing:** Tatiana Eremicheva, Yana Nikonova, Svetlana Alexeeva.

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
