## [Decision Letter · Decision Letter 0]

16 Sep 2025

Dear Dr. Eremicheva,

Thank you for submitting your manuscript to PLOS ONE. After careful consideration, we feel that it has merit but does not fully meet PLOS ONE’s publication criteria as it currently stands. Therefore, we invite you to submit a revised version of the manuscript that addresses the points raised during the review process.

The manuscript presents valuable data on phonological processing and adolescent reading but is considered to require substantial revision to meet PLOS ONE’s publication standards. Both reviews converge on the need for clearer theoretical framing, stronger methodological transparency, and more rigorous statistical reporting. Key concerns include the ceiling effect in the CSP test, the limited validity of the proposed thresholds, insufficient description of materials and procedures, and underdeveloped analyses that do not fully support the claims.

A recommendation of **major revision** is being made. The authors are expected to clarify the theoretical model, reframe normative claims as exploratory, expand methodological detail (including task order, scoring, and inter-rater reliability), and strengthen statistical reporting (β coefficients, confidence intervals, model diagnostics). Claims of clinical or practical significance should be tempered in light of the small effect sizes observed. If these revisions are comprehensively addressed, the manuscript may be suitable for reconsideration.

We look forward to receiving your revised manuscript.

Kind regards,

Ramandeep Kaur

Academic Editor

PLOS ONE

Journal Requirements:

This research was supported by the Basic Research Program at the National Research University Higher School of Economics (HSE University).

4. Please amend the manuscript submission data (via Edit Submission) to include author Yana Nikonova

5. Please amend your authorship list in your manuscript file to include author Yana Nikonona

Additional Editor Comments:

Thank you for submitting your manuscript on phonological processing skills and reading fluency in Russian-speaking adolescents. This is a valuable topic, and your dataset has the potential to make a meaningful contribution. However, substantial revisions are necessary before the manuscript can be considered for publication.

A. Revisions Required for Consideration

Theoretical Framework

Clearly distinguish phonological processing components (e.g., PA, PWM) from the tests used to measure them (e.g., RAN, CSP).

Correctly situate RAN as a task rather than a component.

Explain how CSP maps onto specific PP components and how it differs from RAN in terms of cognitive processes.

Discuss the role of Russian orthographic and morphological features in interpreting the findings.

CSP Thresholds

Reframe the reported thresholds as sample-referenced and exploratory, not normative, given the ceiling effects and small per-grade subsamples.

Justify the choice of a 10% cut-off and provide sensitivity analyses with alternative thresholds.

Methods Transparency

Participants: Describe recruitment procedures and provide justification for inclusion/exclusion criteria. Clarify how gender was analyzed.

Materials: Address similarity between word and pseudoword lists, and potential carryover effects. Provide information about the text-reading passage (genre, curriculum relation).

Procedure: Clarify task order, whether it was counterbalanced, and address possible fatigue or order effects.

Scoring: Provide information on number of raters, their training, inter-rater reliability, and rules for handling errors and self-corrections.

Statistical Analyses and Reporting

Report β coefficients, standard errors, 95% confidence intervals, exact p-values, and adjusted R² values.

Clarify your approach to multiple comparisons.

Explore interactions (e.g., PP × grade, PP × gender) and test for non-linear effects, or justify their omission.

Provide descriptive statistics and distributions for all measures, including checks for ceiling/floor effects.

Interpretation of Findings

Temper conclusions to reflect the modest incremental variance explained (~2% for text fluency).

Discuss limitations posed by the CSP ceiling effect and the limited predictive power for word and pseudoword fluency.

Consider the potential role of comprehension in interpreting text fluency results.

Presentation and Language

Condense the introduction to better highlight the research gap in adolescents.

Introduce technical terms clearly upon first mention.

Correct phrasing such as “words fluency” to “word fluency” throughout.

Standardize reference formatting and ensure completeness.

B. Recommended Improvements

Consider including or at least discussing alternative PP measures (e.g., RAN, spoonerisms) to complement CSP.

Explore the possibility that PP may interact with broader language or cognitive skills (e.g., vocabulary, comprehension).

Enhance visualizations by including distribution plots and gender-specific markers.

Recast “clinical relevance” claims more cautiously, focusing on text-level reading under cognitive load.

Reviewers' comments:

Reviewer's Responses to Questions

**Comments to the Author**

1. Is the manuscript technically sound, and do the data support the conclusions?

Reviewer #1: Yes

Reviewer #2: Partly

2. Has the statistical analysis been performed appropriately and rigorously?

Reviewer #1: Yes

Reviewer #2: No

3. Have the authors made all data underlying the findings in their manuscript fully available?

Reviewer #1: Yes

Reviewer #2: Yes

4. Is the manuscript presented in an intelligible fashion and written in standard English?

Reviewer #1: Yes

Reviewer #2: Yes

Reviewer #1: Reviewer Comments:

Manuscript Title: Phonological processing skills influence text-reading fluency in Russian-speaking adolescents

Decision: Major Revision

Dear Authors,

Thank you for submitting your manuscript titled "Phonological processing skills influence text-reading fluency in Russian-speaking adolescents." Your study addresses an important and underexplored topic by investigating phonological processing (PP) skills and reading fluency in Russian-speaking adolescents. The novelty of examining PP effects simultaneously on words, pseudowords, and text reading, and providing normative thresholds for PP assessment in this population, represents a valuable contribution.

However, several major issues must be addressed before the manuscript can be considered for publication:

1. Ceiling Effect in the PP Test:

* Approximately 75% of participants scored at the top of the CSP test, indicating that this PP test may not be sufficiently challenging for adolescents. Discuss the implications of this limitation and consider additional analyses or alternative tools.

2. Effect Size and Practical Significance:

* The manuscript reports that PP explained only \~2% of variance in text reading fluency and did not significantly predict word or pseudoword reading. This small effect size should be acknowledged, and the practical or clinical implications discussed. Consider addressing language-specific features of Russian that might influence these results.

3. Statistical Analyses:

* The regression models are relatively simple, and Adjusted R² values are low. The manuscript should report additional statistical details (β coefficients, standard errors, confidence intervals) and clarify whether corrections for multiple comparisons were applied.

* Consider exploring interactions, non-linear effects, or multilevel models to better capture potential influences of PP on reading outcomes.

4. Discussion and Interpretation:

* Expand discussion of gender differences observed in word reading but not in other reading measures.

* Clarify the implications of the small PP effect on text reading and discuss limitations of using CSP as the sole PP measure.

* Provide concrete recommendations for future research, including adaptation of other PP assessments (CTOPP, RAN, Spoonerism tasks) for Russian adolescents.

Summary:

*The manuscript titled "Phonological processing skills influence text-reading fluency in Russian-speaking adolescents" presents valuable data and fills an important research gap, but substantial revisions are required to address methodological limitations, clarify findings, and strengthen statistical reporting.

Introduction:

1. Length and Excessive Detail

The introduction is very lengthy and highly detailed. While comprehensiveness is appreciated, in a high-impact journal, introductions are typically concise (3–5 paragraphs) and should lead the reader quickly to the research gap. Sections such as “Phonological processing skills and reading in children” and “Phonological processing skills and reading in adolescents” are overly detailed with study-specific results, which could be summarized more effectively.

2. Structure and Logical Flow

The current structure sometimes reads as a list of prior studies rather than a coherent narrative. For example, after discussing PP in children, the manuscript delves into very specific study details and then moves to adolescents, which disrupts the logical flow. I recommend presenting a general framework for PP and its relevance to reading first, then clearly identifying the research gap in adolescents, and finally stating the study aims.

3. Use of Technical Terms

Specialized terms such as “Dual Route Cascaded model”, PA, PWM, and RAN are introduced without sufficient explanation. While these may be familiar to experts, a brief clarifying sentence for each term upon first introduction would improve accessibility for a broader readership.

4. Lack of Critical Perspective

The introduction mostly summarizes prior studies descriptively but does not critically evaluate their limitations. High-tier journals expect authors to actively highlight gaps in existing literature and justify the significance of the present study.

5. Presentation of Aims and Hypotheses

The study aims and hypotheses are presented only at the end of the introduction, after extensive details. These could be stated earlier and more clearly to orient the reader to the purpose of the research from the beginning.

6. References and Citations

Some references appear incomplete or inconsistently formatted (e.g., \[36]). The citation list and in-text references should be carefully checked for accuracy and consistency.

7. Language and Style

Sentences are often long and complex, sometimes containing multiple ideas. This affects readability. Shorter, clearer sentences with one main idea each would improve clarity and reader engagement.

8. Limited discussion of language-specific characteristics

While the study focuses on Russian-speaking adolescents, the introduction does not adequately address structural and phonological features of Russian that may influence phonological processing (PP) and reading. Differences in orthographic transparency, phoneme-grapheme correspondence, and reading norms should be highlighted to justify the relevance of studying this population.

9. Insufficient integration of broader language and cognitive skills

The introduction frames PP primarily as a predictor of reading, but reading in adolescents involves interaction with other language and cognitive skills (e.g., grammar, vocabulary, text comprehension). A more comprehensive framework linking PP to these broader skills would strengthen the rationale for the study.

10. Limited discussion of cognitive complexity in adolescent reading

While detailed for children, the introduction provides less explanation of the additional cognitive demands of reading in adolescents, such as processing longer and more complex texts. This is critical for understanding why PP alone may not fully predict reading outcomes in this age group.

11. Insufficient justification of the assessment instruments

Although RuToPP, STARS, and SARS are mentioned, the introduction does not clearly specify which tasks target PA, PWM, or other components of PP, nor the validity of these instruments for Russian-speaking adolescents. Clear justification for the choice of instruments is needed.

12. Lack of a coherent theoretical framework for adolescents

The traditional three-component PP model (PA, PWM, RAN) is referenced, but the introduction does not critically evaluate its applicability to adolescents or discuss potential limitations or modifications needed for this age group.

13. Limited discussion of practical or clinical implications

The introduction emphasizes theoretical significance but gives limited attention to potential applications in identifying reading difficulties or informing intervention programs for adolescents. Explicitly linking the study to practical outcomes would increase its relevance.

**Summary Recommendation**

Overall, the introduction provides a thorough overview, but it would benefit from condensation, clearer structure, stronger critical framing of prior work, and more prominent presentation of study aims and hypotheses. These revisions will help engage readers and strengthen the rationale for your study.

The introduction would benefit from a stronger focus on language-specific considerations, integration of broader linguistic and cognitive skills, justification of the assessment tools, and explicit discussion of theoretical and practical implications for adolescent reading. Addressing these points would enhance both the scientific rigor and applied relevance of the study.

Methods:

1. Participants

* The sample size (N=163) is adequate, but the sampling procedure is not fully described. Were schools randomly selected, or was participation voluntary? This affects the generalizability of the findings.

* Gender distribution is reported, but the approach to analyzing potential gender differences in reading or phonological skills is not clearly described.

* The inclusion criterion of nonverbal intelligence is reasonable, but the rationale for excluding participants solely based on below-norm scores should be more transparent.

2. Materials – Words and Pseudowords Reading Test

* The STARS test is adequately described, but the selection of lists C and D requires further justification. Why were only these lists used instead of counterbalancing across participants? This could lead to order or list effects.

* Stimulus complexity is described in terms of frequency and syllable count, but additional information about phonotactic legality and morphological complexity would strengthen the assessment of appropriateness for adolescents.

3. Text Reading Test

* Using a novel text (“The Caring Flower”) to avoid prior familiarity is reasonable, but the standardization is deferred to a future article. Brief information on reliability and validity should be included here to ensure interpretability.

* Comprehension questions include open- and closed-ended formats, but scoring procedures for open-ended responses and inter-rater reliability are not reported.

4. Phonological Processing Test (CSP from RuToPP)

* The selection of CSP as the most complex test is justified, but confirmation that ceiling effects are not present in adolescents is needed. Was a pilot study conducted?

* The connection between the five speech processes and theoretical PP components (PA, PWM, RAN) should be made explicit.

* Only a single PP measure is used, which limits the ability to examine different facets of phonological processing independently.

5. Procedure

* Task order is clear, but potential fatigue or order effects are not addressed. CSP was administered last; prior tasks may influence performance.

* Scoring methods for reading (handling self-corrections and errors) should be described in detail.

6. Data Analysis

* Linear regression and logarithmic transformation are appropriate.

* Further clarification is needed regarding coding of gender and treatment of grade as a numeric variable.

* Non-linear relationships between age/grade and reading/PP may be relevant and should be considered.

* Reporting effect sizes and confidence intervals would enhance interpretability.

7. Clarity and Reproducibility

* Overall methods are detailed, but information on text standardization, comprehension scoring, task order, and scorer reliability should be expanded for full reproducibility.

Summary Recommendation

The methods are thorough and well-organized. Authors should:

1. Provide additional justification for instrument selection and task order,

2. Include brief reliability and validity information for the text reading task,

3. Address potential ceiling, fatigue, and order effects in adolescents,

4. Expand statistical analysis reporting, including effect sizes and potential non-linear relationships,

5. Clarify scoring procedures and inter-rater reliability for reading and phonological tasks.

Results:

1. Descriptive statistics and data distribution

* The authors correctly assessed the CSP score distribution and appropriately used percentile-based thresholds due to non-normality.

* Presenting both percentile-based and mean ± SD thresholds may be somewhat confusing. It would be clearer to select one primary metric for analysis and report the other as supplementary information.

2. Regression models and variance explained

* Regression models for words, pseudowords, and text are appropriate, but the Adjusted R² values are very low (0.041–0.099), indicating that PP explains only a small portion of reading variance. This should be explicitly discussed to avoid overinterpretation.

* For models where PP did not significantly predict word or pseudoword reading, additional analyses (e.g., non-linear models, interaction terms, or multilevel models) could be considered to determine whether small effects might emerge under more complex modeling.

3. Effects of gender and grade

* A significant gender effect is observed for word reading (females faster), but not for pseudowords or text reading. This discrepancy should be discussed, considering possible explanations such as scoring methods, motivation, or linguistic factors.

* Grade level does not have a significant effect, but potential non-linear developmental changes across grades 8–11 should be considered.

4. Interpretation of PP effect on text reading

* PP significantly predicts text reading, but the additional variance explained is only \~2%. This is a very small effect and should be interpreted cautiously.

* Discussion should consider the practical or clinical relevance of this small effect.

5. Clarity and reporting of data

* References to figures and tables are clear, but more detailed statistics (e.g., β coefficients, SE) would improve interpretability.

* Clarification on multiple comparison corrections or p-value adjustments, if any, should be provided.

6. Overall summary

* Results are systematically and clearly presented. However, the effect of PP on reading is limited and small, and some models do not reach significance. The discussion should explicitly address these limitations and the small effect sizes, as well as developmental considerations.

Discussion:

1. Innovation and study significance

* The authors appropriately highlight the novelty of simultaneously examining PP effects on words, pseudowords, and text reading.

* However, claims regarding clinical relevance should be cautious, as PP explains only \~2% of text reading variance, which may limit practical applications.

2. PP thresholds and practical application

* Providing percentile-based normative thresholds for CSP is valuable for identifying potential phonological difficulties.

* Limitations and generalizability of these thresholds to other populations (e.g., different geographic or socio-economic groups) should be discussed.

3. PP and text-level reading

* The discussion of PP effects on text reading and similarity to RAN is useful.

* The small effect size and cross-linguistic differences warrant more emphasis, including the practical significance of 2% variance explained.

* Comparison with English-language studies is logical, but a more detailed analysis of Russian orthography features (phoneme transparency, number of phonemes, morphological complexity) would strengthen interpretation.

4. PP and word-level reading

* Reporting the lack of PP effect on words and pseudowords is clear, but the discussion could include deeper analysis of potential interactions with other cognitive or linguistic skills beyond lexical access.

5. Gender differences

* The finding that females read words faster is interesting, while pseudoword and text reading show no gender differences.

* While supported by prior literature, limitations related to sample size, cultural factors, and orthographic transparency of Russian should be acknowledged.

6. Limitations and future directions

* Main limitations, including small effect sizes and limited PP measures, should be emphasized.

* Future research suggestions could include using complementary RAN measures, examining interactions of PP with vocabulary or comprehension skills, and longitudinal reading assessment.

Limitations and Conclusion:

1. Limitations of the PP tool

* The authors correctly acknowledge a ceiling effect; approximately 75% of participants achieved the highest CSP score, indicating the test may not be sufficiently challenging for adolescents.

* Detailed analysis of specific items (e.g., palatalized consonants) is valuable, showing that some phonological skills are still developing.

* The recommendation to adapt or develop other tools (CTOPP, RAN, Spoonerism) is appropriate, but potential limitations in standardizing these tools for Russian should be discussed further.

2. Limitations related to word-level reading

* The analysis indicates that PP does not significantly predict word-level reading, with plausible explanations including sample size, test sensitivity, or developmental ceiling.

* The discussion correctly argues that sample size is likely not the main limitation, aligning with findings from larger studies.

* Concluding that PP is no longer a significant predictor for word-level reading in adolescents is consistent with previous research.

3. Implications for text-level reading

* Findings suggest that PP can still influence text reading in adolescents, but the effect is small and may reflect overall cognitive load rather than specific phonological skills.

* The suggestion to investigate cognitive load in future research is well-founded, though the small effect size and practical relevance should be emphasized.

4. Clarity and future directions

* Limitations and conclusions are clearly presented, but more concrete recommendations for future research (e.g., developing more challenging tests for older adolescents) would strengthen the discussion.

Summary:

* Main limitation: ceiling effect and test sensitivity.

* PP affects text-level reading, but the effect is limited and may reflect overall cognitive load.

* Findings are consistent with the literature, and future research suggestions regarding tool development and cognitive load assessment are appropriate.

Reviewer #2: Review of Phonological processing skills influence text-reading fluency in Russian-speaking adolescents

Summary: The manuscript describes a study in which phonological processing skills, word-, pseudoword and text reading fluency was studied in four age-groups of adolescents. The results showed that a test of phonological processing skills significantly predicted text reading fluency.

General evaluation: The introduction of the study convinced me that the research fills an important gap in focusing on how adolescents reading skills might be still determined by their phonological skills. However, I think the sample size, especially the group sizes are not suitable for providing normative data on the used PP test. I also do not agree of the author’s interpretation of the results when they compare the three regression models. The result and the method section could be improved by providing more information. The text also contains a few language mistakes which need to be corrected. The study is not appropriate for publication without major revisions.

Review in details:

Introduction:

1) The introduction strongly focuses on which PP tests were found to be associated with different reading tests which is indeed an important question, but a clearer explanation of the processing during reading and how the different tests tap into these processes is missing. For example, the traditional model of PP (page 4) should be explained in a way that a difference is made between the components and the tests used to measure it, it should be explained how the different components relate to reading. For example, the RAN does not seem to be a component of any model (page 4), but a method to measure certain components of reading. Please, also correct this on page 7: “The third component of the traditional PP model, i.e., RAN…”.

2) On page 9, it is not clear whether the chosen PP test only involves PA and PWM components or it might also require other processes. Moreover, I find it a bit controversial that somehow the introduction argues that the use of RAN test in other studies is a bit problematic because the RAN is a complex task, hence, it is less clear which processes predict reading skills when an association is found between reading fluency/skills and RAN. Then, the authors choose a PP test for the present study which is also complex. Hence, I wonder what do we gain from choosing the CSP test and not the RAN? Is it possible to explain the differences between the two tests in terms of underlying cognitive and linguistic processing?

Methods:

3) Please, indicate whether the text used in the Text reading test was expository or narrative and whether it was part of school curriculum.

4) On page 12, it is stated that “The list D included pseudowords which were created by taking existing Russian words from list C and replacing 1-3 graphemes in them.” This suggest to me that the words in two lists, measuring word and pseudo word fluency were similar. Is it so? And if yes, could similarities between the words of the two lists influence the results of the list which was given to the students as second? And a related question: what was the order of the words and pseudoword tests? This is not clear from the Procedure section on page 14. Which one was presented first? Was the order changes across participants? All participants got the same list of words for the two tasks?

5) Test evaluations: If I understand well performance of all tests needed to be evaluated by the researchers to get the final scores. Please, provide information on how many persons were involved in the scoring of the results. Inter-rater reliability should be provided (at least) on some part of the data of each task.

6) Data analysis:

On page 16, the authors write that “The dependent, within-subject variables were words, pseudowords and text reading fluency”. As three independent regression models were run on the three response variables, i.e. the variables were not included in one statistical model, these are not really within-subject variables, just three different response variables from the same sample. Reading “within-subject variable” was a bit confusing when I tried to understand the description of the data-analysis.

Results

7) Threshold for phonological test:

In general, a group of data with sample size between 31 and 55 is too low and not representative for any normalization and thresholding, especially, if it is not done together with other measures (e.g., clinical assessment of reading and language difficulties) which could reliably show that low performing participants on certain test have reading, and language difficulties confirmed by other sources as well. It is also not clear why 10% was chosen as a threshold for risk.

8) There was a significant gender difference in word reading fluency but the Figure 2 does not show data colored/indicated by gender. Gender information should be added to Figure 2 or it should presented in the separate figure.

9)There is no information on the descriptive statistics of any of the fluency task, although it would be important to see whether it is possible that there is any ceiling effect or if there might be differences in the variation between these variables.

10) I do not find information whether the results of the comprehension test have been used for assessing validity of the text fluency test, and whether there was any tradeoff between fluency and comprehension. This would be important in the interpretation of the association between CSP test results and text reading fluency.

Discussion

10) In general, I am not convinced that the provided threshold for the CSP test (see above) could be used to identify the potential risk of phonological deficit in adolescents.

11) On page 24, the authors give some reasons for the lack of the effect of the PP skills on the word-level fluency. I would also add that the word-fluence tasks might be less sensitive, i.e. there might be less variation in the word-fluency measures compared to the text fluency to show any reliable effect (i.e. if the majority of the participants actually easily could do the task, there is no variation to predict). To rule out this explanation, please, also add the information of the distribution of these variables (see my point above).

Conclusion

12) On page 26, the authors write: “the complex structure of the PP test suggest that it is not specific phonological skills, but rather the overall level of cognitive load that may significantly influence adolescents’ reading fluency”.

It is not clear to me why the authors rule out that some phonological processing skills measured by the PP test could be associated with text reading fluency. Is it because this effect was not found on the word- and pseudo-word level? In this case, please, write that also in the conclusion. Although I am still not convinced about strong conclusions could be made about the lack of effect on the word- and pseudo-word level, because of the lack of information about the results of these tests (see my point above). However, in any case, it would be nice to read more discussion on what are the underlying processes in the used PP test (CSP), and how those could influence reading.

13) On page 26 (last sentence): “Therefore, further studies might be devoted to the assessment of cognitive load during PP and other cognitive tests in adolescents and its impact on reading texts.” It is not “the cognitive load during PP or the cognitive tests” which impact reading texts, but the cognitive load in general. Please, rephrase this sentence.

Language in general

14) I am not a native speaker of English, therefore, I will not comment on language in detail but it appeared to me that it is “word fluency” and not “words fluency”. Please, correct it throughout the paper.

**Do you want your identity to be public for this peer review?** For information about this choice, including consent withdrawal, please see our Privacy Policy

Reviewer #1: No

Reviewer #2: No

---

## [Author Response · Author response to Decision Letter 1]

10 Nov 2025

Journal Requirements:

https://journals.plos.org/plosone/s/file?id=wjVg/PLOSOne_formatting_sample_main_body.pdfand

Answer: We ensured the journal’s requirements and revised our file naming.

This research was supported by the Basic Research Program at the National Research University Higher School of Economics (HSE University).

Answer: The funders had no role in study design, data collection and analysis, decision to publish, or preparation of the manuscript.

Answer: Thank you for your concern! We updated the information on data acceptance. Our data is now freely accessible via the OSF platform: https://osf.io/p6f24/overview?view_only=de243c43b160419783d68a5f171db6f5

4. Please amend the manuscript submission data (via Edit Submission) to include author Yana Nikonova

Answer: Thank you for your comment! We included author Yana Nikonova in the submission data.

5. Please amend your authorship list in your manuscript file to include author Yana Nikonona

Answer: Thank you for your comment! We included author Yana Nikonova in the manuscript file.

Answer: Thank you for your comment! We did not receive any recommendations on the specific works that we need to cite in our manuscript.

Additional Editor Comments:

Thank you for submitting your manuscript on phonological processing skills and reading fluency in Russian-speaking adolescents. This is a valuable topic, and your dataset has the potential to make a meaningful contribution. However, substantial revisions are necessary before the manuscript can be considered for publication.

Answer to the editor: We sincerely thank you for your valuable comments and the opportunity to revise our manuscript. We carefully considered all the feedback provided and made substantial revisions to improve the clarity, structure, and overall quality of the paper.

A. Revisions Required for Consideration

Theoretical Framework

Clearly distinguish phonological processing components (e.g., PA, PWM) from the tests used to measure them (e.g., RAN, CSP).

Correctly situate RAN as a task rather than a component.

Answer: Thank you for the valuable comments on the features of the traditional PP model. We corrected the usage of the relevant terminology throughout the manuscript and accurately proposed the model’s components in the Introduction section (see page 4).

Explain how CSP maps onto specific PP components and how it differs from RAN in terms of cognitive processes.

Answer: We explained the features of the CSP test and its relation to the traditional PP model in the Introduction section (the paragraph devoted to This Study; see page 11).

Discuss the role of Russian orthographic and morphological features in interpreting the findings.

Answer: Thank you for the relevant suggestions on the role of Russian grammar in interpreting the results. The more detailed description can be seen in the Introduction section (the paragraph devoted to This Study; see pages 13-14).

CSP Thresholds

Reframe the reported thresholds as sample-referenced and exploratory, not normative, given the ceiling effects and small per-grade subsamples.

Answer: Thank you for the suggestions on the CSP thresholds. We corrected the reported information according to your comments. See the Results section (page 27-28) and the Discussion section (pages 32-33).

Justify the choice of a 10% cut-off and provide sensitivity analyses with alternative thresholds.

Answer: We thank the reviewer for this valuable comment. The rationale for selecting the 10% cut-off is now explained in the revised manuscript (page 27, paragraph Thresholds for phonological test).

Methods Transparency

Participants: Describe recruitment procedures and provide justification for inclusion/exclusion criteria.

Answer: We thank the reviewer for this comment. The recruitment procedures and inclusion/exclusion criteria are now described in detail in the revised manuscript (pages 14-16, paragraph Participants).

Clarify how gender was analyzed.

Answer: We appreciate the reviewer’s suggestion. The approach to analyzing gender is now clarified in the revised manuscript (pages 24-26, paragraph Data analysis).

Materials: Address similarity between word and pseudoword lists, and potential carryover effects.

Answer: Thank you for this important comment. We have addressed the similarity of word and pseudoword lists and discussed potential carryover effects in the manuscript (pages 16-17, paragraph Word and pseudoword reading tests).

Provide information about the text-reading passage (genre, curriculum relation).

Answer: We appreciate the reviewer’s attention to detail. Information on the text-reading passage, including its genre and relation to the curriculum, has been added (pages 17-18, paragraph Text reading test).

Procedure: Clarify task order, whether it was counterbalanced, and address possible fatigue or order effects.

Answer: Thank you for this comment. Task order, counterbalancing, and potential fatigue and order effects are now clarified in the revised manuscript (pages 21-24, section Procedure; pages 41-42, paragraph Limitations of our study and directions for further research).

Scoring: Provide information on number of raters, their training, inter-rater reliability, and rules for handling errors and self-corrections.

Answer: Thank you for the relevant comments on the scoring procedure! We decided to add a separate paragraph dedicated to the details of scoring in this study (see pages 23-24, paragraph Manual scoring)

Statistical Analyses and Reporting

Report β coefficients, standard errors, 95% confidence intervals, exact p-values, and adjusted R² values.

Answer: We thank the reviewer for this suggestion. We have now included a table reporting β coefficients, standard errors, 95% confidence intervals, exact p-values, and adjusted R² values for all three final models in the revised manuscript (Results section, Table 3).

Clarify your approach to multiple comparisons.

Answer: We appreciate the reviewer’s comment and acknowledge the importance of addressing multiple comparisons. In our study, we did not apply corrections for multiple comparisons, as we conducted three independent analyses. Word and pseudoword reading fluency cover two independent ways (sublexical and lexical, respectively) in the dual-route model (Coltheart et al., 2001); therefore, it is unlikely that they influence each other.

Regarding text reading fluency, although text reading engages mechanisms involved in both word and pseudoword reading (as readers may occasionally encounter unfamiliar words), we consider text reading fluency to be an independent variable. Text reading involves contextual and predictive processes that substantially change reading behavior compared to reading isolated words or pseudowords. Therefore, we treated these analyses as separate rather than multiple comparisons requiring correction.

Explore interactions (e.g., PP × grade, PP × gender) and test for non-linear effects, or justify their omission.

Answer: We thank the reviewer for this comment. Interactions (e.g., PP × grade, PP × gender) have been added to the models, and the results are now described in the Results section (pages 27-31). Regarding non-linear effects, we consider their inclusion unnecessary because most predictors do not have sufficient unique observations to support such analyses. The only predictor with a sufficiently large number of data is the IQ score, which is reasonably approximated as linear.

Provide descriptive statistics and distributions for all measures, including checks for ceiling/floor effects.

Answer: Thank you for this suggestion. Descriptive statistics and distributions for all measures are provided in the supplementary materials (S1-S4 Figs, S5 Table).

Interpretation of Findings

Temper conclusions to reflect the modest incremental variance explained (~2% for text fluency).

Answer: Thank you for the valuable comments on our manuscript! We considered your recommendations and provided more tempered conclusions on the effect size in the Discussion section of this manuscript (see pages 33-34, paragraph Phonological test and text-level reading; pages 40-41).

Discuss limitations posed by the CSP ceiling effect and the limited predictive power for word and pseudoword fluency.

Answer: Thank you for the valuable comments on our manuscript! We considered your recommendations and provided a more detailed discussion on the CSP ceiling effect in the Discussion section of this manuscript (see pages 38-40, paragraph Limitations of our study and directions for further research). The predictive power of word and pseudoword reading tests is discussed on pages 34-37 (paragraph Phonological test and word-level reading).

Consider the potential role of comprehension in interpreting text fluency results.

Answer: We thank the reviewer for this suggestion. We added comprehension question scores as a predictor in the model predicting text reading fluency; however, this variable did not reach statistical significance, so it wasn’t included in the final model (see the Results section, pages 28-31).

Presentation and Language

Condense the introduction to better highlight the research gap in adolescents.

Answer: Thank you for the valuable comments on our manuscript! We considered your recommendations and provided major revisions to the Introduction section of this manuscript (see pages 3-14).

Introduce technical terms clearly upon first mention.

Answer: Thank you for the comment! We considered your recommendation and provided more detail on the technical terms used in our manuscript.

Correct phrasing such as “words fluency” to “word fluency” throughout.

Answer: Thank you for the recommendation! We edited this term throughout the text of our manuscript.

Standardize reference formatting and ensure completeness.

Answer: Thank you for the valuable comments on our manuscript! We checked the reference style in our manuscript.

B. Recommended Improvements

Consider including or at least discussing alternative PP measures (e.g., RAN, spoonerisms) to complement CSP.

Answer: Thank you for the valuable comment on the alternative PP measures. We discussed this topic in more detail in the Discussion section of this manuscript (pages 39-41, paragraph Limitations of our study and directions for further research).

Explore the possibility that PP may interact with broader language or cognitive skills (e.g., vocabulary, comprehension).

Answer: We thank the reviewer for this suggestion. Based on the available data, we examined interactions of PP with participant’s grade, gender, CSP test scores, Raven’s Standard Progressive Matrices scores, total memory scores, reading habits index, and comprehension question scores. The results of these analyses are now reported in the revised manuscript (see Data analysis and Results sections, pages 24-26, 28-31).

Enhance visualizations by including distribution plots and gender-specific markers.

Answer: Thank you for this comment. Results, including distributions by gender, are presented in Fig. 2.

Recast “clinical relevance” claims more cautiously, focusing on text-level reading under cognitive load.

Answer: Thank you for the commentary on the relationship between PP and reading tests. We reframed the part on the cognitive load in the Discussion section (see pages 42-43).

Reviewer #1: Reviewer Comments:

Manuscript Title: Phonological processing skills influence text-reading fluency in Russian-speaking adolescents

Decision: Major Revision

Dear Authors,

Thank you for submitting your manuscript titled "Phonological processing skills influence text-reading fluency in Russian-speaking adolescents." Your study addresses an important and underexplored topic by investigating phonological processing (PP) skills and reading fluency in Russian-speaking adolescents. The novelty of examining PP effects simultaneously on words, pseudowords, and text reading, and providing normative thresholds for PP assessment in this population, represents a valuable contribution.

However, several major issues must be addressed before the manuscript can be considered for publication:

General response: We are sincerely grateful for your exceptionally detailed and thoughtful review. We truly appreciate your careful attention to our work and the valuable feedback. We have carefully considered and addressed all your comments and suggestions to the best of our ability.

1. Ceiling Effect in the PP Test:

* Approximately 75% of participants scored at the top of the CSP test, indicating that this PP test may not be sufficiently challenging for adolescents. Discuss the implications of this limitation and consider additional analyses or alternative tools.

Answer: Thank you for the valuable comments on our manuscript! We considered your recommendations and discussed the ceiling effect of the CSP test in the Discussion section of this manuscript (see pages 38-40, paragraph Limitations of our study and directions for further research).

2. Effect Size and Practical Significance:

* The manuscript reports that PP explained only \~2% of variance in text reading fluency and did not significantly predict word or pseudoword reading. This small effect size should be acknowledged, and the practical or clinical implications discussed. Consider addressing language-specific features of Russian that might influence these results.

Answer: Thank you for the valuable comments on our manuscript! We considered your recommendations and provided more tempered conclusions on the effect size in the Discussion section of this manuscript (see pages 33-34, paragraph Phonological test and text-level reading; pages 40-41). We also discussed language-specific features of Russian in more detail (see page 40-41).

3. Statistical Analyses:

* The regression models are relatively simple, and Adjusted R² values are low. The manuscript should report additional statistical details (β coefficients, standard errors, confidence intervals) and clarify whether corrections for multiple comparisons were applied.

* Consider exploring interactions, non-linear effects, or multilevel models to better capture potential influences of PP on reading outcomes.

An

---

## [Editor Report · Decision Letter 1]

12 Nov 2025

Phonological processing skills influence text-reading fluency in Russian-speaking adolescents

PONE-D-25-32670R1

Dear Dr. Eremicheva,

We’re pleased to inform you that your manuscript has been judged scientifically suitable for publication and will be formally accepted for publication once it meets all outstanding technical requirements.

Kind regards,

Ramandeep Kaur

Academic Editor

PLOS ONE
---

## [Editor Report · Acceptance letter]

PONE-D-25-32670R1

PLOS ONE

Dear Dr. Eremicheva,

I'm pleased to inform you that your manuscript has been deemed suitable for publication in PLOS ONE. Congratulations! Your manuscript is now being handed over to our production team.

Kind regards,

on behalf of

Dr. Ramandeep Kaur

Academic Editor

PLOS ONE